

# Predator in proximity: how does a large carnivore respond to anthropogenic pressures at fine-scales? Implications for interface area management

Manu Mohan[1,2], Sambandam Sathyakumar[3] and Ramesh Krishnamurthy[2,4]

[1] Post-Graduate Programme in Wildlife Science, Wildlife Institute of India, Dehradun, Uttarakhand, India
[2] Department of Landscape Level Planning and Management, Wildlife Institute of India, Dehradun, Uttarakhand, India
[3] Department of Endangered Species Management, Wildlife Institute of India, Dehradun, Uttarakhand, India
[4] Faculty of Forestry, University of British Columbia, Vancouver, British Columbia, Canada

Corresponding author
Ramesh Krishnamurthy,
ramesh@wii.gov.in

## ABSTRACT

**Background:** Driven by habitat loss and fragmentation, large carnivores are increasingly navigating human-dominated landscapes, where their activity is restricted and their behaviour altered. This movement, however, raises significant concerns and costs for people living nearby. While intricately linked, studies often isolate human and carnivore impacts, hindering effective management efforts. Hence, in this study, we brought these two into a common framework, focusing on an interface area between the critical tiger habitat and the human-dominated multiple-use buffer area of a central Indian protected area.

**Methods:** We employed a fine-scale camera trap survey complemented by GPS-collar movement data to understand spatio-temporal activity patterns and adjustments of tigers in response to anthropogenic pressures. We used an occupancy framework to evaluate space use, Bayesian circular GLMs to model temporal activity, and home range and step length analyses to assess the movement patterns of tigers. Further, we used predation-risk models to understand conflict patterns as a function of tiger presence and other habitat variables.

**Results:** Despite disturbance, a high proportion of the sampled area was occupied by 17 unique tigers ($\psi$ = 0.76; CI [0.73–0.92]). The distance to villages ($\beta \pm$ SE = 0.63 $\pm$ 0.21) and the relative abundance of large-bodied wild prey ($\beta \pm$ SE = 0.72 $\pm$ 0.37) emerged as key predictors of tiger space use probability, indicating a preference for wild prey by tigers, while human influences constrained their habitat utilisation. Distance to villages was also identified as the most significant predictor of the tigers' temporal activity ($\mu \pm \sigma$ = 3.03 $\pm$ 0.06 rad) that exhibited higher nocturnality near villages. A total of 11% of tiger home ranges were within village boundaries, accompanied by faster movement in these areas (displacement 40–82% higher). Livestock depredation probability by tigers increased with proximity to villages ($P$ = 0.002) and highway ($P$ = 0.003). Although tiger space use probability ($P$ = 0.056) and wild prey abundance ($P$ = 0.134) were non-significant at the 0.05 threshold, their presence in the best-fit predation-risk model suggests their contextual relevance for

understanding conflict risk. The results highlight the importance of appropriately managing livestock near human infrastructures to effectively mitigate conflict.

**Conclusions:** Shared space of carnivores and humans requires dynamic site-specific actions grounded in evidence-based decision-making. This study emphasises the importance of concurrently addressing the intricate interactions between humans and large carnivores, particularly the latter's behavioural adaptations and role in conflict dynamics. Such an integrated approach is essential to unravel cause-effect relationships and promote effective interface management in human-dominated landscapes.

# INTRODUCTION

With the rise in anthropogenic pressures across the globe, conflicts with wild animals have become more common in the recent past (*König et al., 2020*; *Bombieri et al., 2023*). Large carnivores are one such extremely vulnerable group of animals that are subject to direct persecution, habitat loss, and fragmentation, making them most vulnerable to space-dependent extinction (*Sillero-Zubiri & Laurenson, 2001*; *Ripple et al., 2014*; *Hirt et al., 2021*). Such conditions cause them to venture into human-dominated areas in search of prey and new territories (*Ripple et al., 2014*), leading to conflict with humans, acting as the single-most-important cause of mortality for large carnivores (*Woodroffe & Ginsberg, 1998*; *Woodroffe, 2000*). Currently, even protected areas (PAs) are under intense anthropogenic stress (*Jones et al., 2018*; *Williams, Rondinini & Tilman, 2022*), which has made the interface between human-dominated and PAs prone to the highest degree of conflict (*Nyhus & Tilson, 2004*; *Karanth & Kudalkar, 2017*). As a result, humans also face a spectrum of direct and indirect losses (*Barua, Bhagwat & Jadhav, 2013*), with significant impacts on their economic well-being (*Dickman, Macdonald & Macdonald, 2011*; *Braczkowski et al., 2023*)

Depending on the risks and resources present, animals adapt to external stressors differently with regard to their space use and activity (*Tuomainen & Candolin, 2011*; *Lamb et al., 2020*). Evidence of spatial and temporal avoidance of humans (*Valeix et al., 2012*; *Oriol-Cotterill et al., 2015a*) and 'nocturnal refuge' adopted by mammals, including apex carnivores, is well-established worldwide (*Gaynor et al., 2018*; *Lamb et al., 2020*). A variety of biophysical variables affect how a large carnivore behaves with respect to its movement and choice of resources in human-dominated areas (*Miller & Schmitz, 2019*; *Wilkinson et al., 2020*), proximity to human infrastructure being a crucial factor (*Soto-Shoender & Giuliano, 2011*; *Soh et al., 2014*; *Miller, Jhala & Jena, 2016*; *Farmer et al., 2022*). Long-term studies have shown that large carnivores face a higher risk of mortality and reduced survival due to the presence of human infrastructures (*Nisi et al., 2023*), even when

protected (*Benson et al., 2023*). In contrast, large carnivores that acclimate to disturbance may turn into bold individuals (*Tuomainen & Candolin, 2011*; *Bombieri et al., 2021*) and have a higher chance of coming into conflict and direct competition with humans (*Johansson et al., 2016*). Understanding such patterns and causal mechanisms is imperative to reduce losses to humans and wildlife. The concept of landscape of coexistence (*Oriol-Cotterill et al., 2015b*) offers a promising approach to exploring and explaining the influence of human-dominated landscapes on large carnivores (*Carter et al., 2012*; *Gehr et al., 2017*; *Lamb et al., 2020*).

Despite conflicts and habitat shrinkages, a tenth of the global large carnivore population has shown signs of recovery (*Ingeman et al., 2022*). In India, the conservation and recovery of tigers is one such example, owing to proactive management (*Jackson, 2010*; *Jhala et al., 2021*; *Qureshi et al., 2023*), including successful conservation programmes, such as reintroduction (*Sankar et al., 2010*; *Ramesh et al., 2015*; *Sarkar et al., 2016*). However, there exists resistance to tiger dispersal in landscapes, often due to human influence (*Joshi et al., 2013*; *Krishnamurthy et al., 2016*; *Reddy et al., 2017*). The consequences are two-fold: one that of the hindrance to the carnivore's ability for dispersal, crucial for their long-term survival (*Wikramanayake et al., 2011*; *Thatte et al., 2018*) and the other of increasing instances of negative encounters with humans around tiger source populations of PAs (*Dhanwatey et al., 2013*; *Miller, Jhala & Jena, 2016*; *Malviya & Krishnamurthy, 2022*).

As tiger populations increase in response to successful conservation measures in tiger reserves across the country, the potential for conflict with adjacent human populations may escalate (*Karanth & Gopal, 2005*). Human-tiger conflict most commonly involves tigers preying on domestic livestock (*Goodrich, 2010*). Among the carnivores in the country, the economic costs of negative interaction with tigers are almost thrice that of leopards and a hundred times that of wolves (*Gulati et al., 2021*), with some parts of the country experiencing losses specifically due to livestock depredation to be at least seven times higher than those caused by leopards (*Malviya & Ramesh, 2015*). *Braczkowski et al. (2023)* estimate that depredation events by tigers in their habitat ranges account for a mean annual per-capita income loss as high as 34% for livestock owners. This estimate shows at least a two-fold apparent increase compared to the results of previous studies in India within the last two-decade span (*Madhusudan, 2003*; *Harihar, Ghosh-Harihar & MacMillan, 2014*). Additionally, 70% of the tiger habitat range falls within areas of high-severe economic vulnerability (*Braczkowski et al., 2023*).

The central Indian landscape, with its burgeoning tiger population (*Qureshi et al., 2023*) and substantial compensation disbursement due to depredation (*Karanth & Kudalkar, 2017*), provides a unique context to investigate the reciprocal impacts between tiger space use and human-wildlife conflict. While previous studies in the landscape have assessed patterns of large-carnivore conflict spatially (*Dhanwatey et al., 2013*), mapped hotspots (*Miller, Jhala & Jena, 2016*; *Malviya & Krishnamurthy, 2022*), integrating social dimensions to assess predation risk (*Karanth et al., 2012*; *Neelakantan, DeFries & Krishnamurthy, 2019*), there exists an unexplored dimension−the concurrent examination of the animals' activity adjustments and their role in conflict occurrences. Such an integrated approach offers a unique opportunity to disentangle cause-effect relationships

underlying the conflict, which enables us to elucidate whether tigers actively avoid human influences or, conversely, if their movements predispose them to encounter livestock, thus exacerbating conflict. This perspective challenges the general perception that tigers primarily target large and easy prey like livestock. Hence, our study introduces a novel perspective to the growing body of literature on tiger conflict and coexistence at fine scales (*Carter et al., 2012*; *Miller et al., 2015*), contributing to the pressing requirement for implementing location-specific management interventions for tiger conservation (*Jhala et al., 2021*). Furthermore, by adopting a fine-scale approach, this study can provide crucial insights for delineating 'protected area-specific' Eco-Sensitive Zones (ESZs, as mandated by the Government of India, *TH, 2023*; *SCI, 2023*), or Zone of Influence, empowering land managers for informed decision-making.

Hence, the present study aims to assess three key aspects in the interface area of a critical tiger habitat (CTH): (i) the spatio-temporal patterns of tiger activity (ii) their adjustments in response to anthropogenic influences, and (iii) their potential contribution to conflict with humans in the form of livestock depredation. We begin by determining the proportion of interface area occupied by tigers and the factors affecting their probability of space use. We hypothesised that there would be a shift in tigers' spatial and temporal activity along an anthropogenic disturbance gradient, such that tigers would be less likely to use areas near to human settlements. We also expected the tigers to be more nocturnal and move faster near villages to avoid human encounters. Finally, we assessed the risk of livestock depredation within the area as a function of tiger presence, along with other habitat variables. We hypothesised that tigers' probability or intensity of space use would significantly contribute to the depredation probability. We conclude by discussing the empirical reasons and consequences of the large carnivore's behavioural shifts and the implications for conflict management, emphasising the importance of understanding and addressing such dynamics in interface areas.

# MATERIALS AND METHODS

## Research approval, permission and ethics statement

The research was reviewed and approved by the Faculty of Wildlife Sciences as part of the Post-graduate Programme in Wildlife Science at the Wildlife Institute of India, Dehradun (No. WII/AC/2019-21/Dissertation/XVII M.Sc, dated 24/11/2020). All necessary permission for conducting the research, including camera trap sampling and other on-ground fieldwork within the PA, was granted by the Madhya Pradesh Forest Department (letter issued by the PCCF (Wildlife)/GEN./15-20/6043, dated 01/09/2018) as part of an ongoing larger landscape project dealing with tiger reintroduction and landscape management (*WII, 2022*). In the study area, human encounters mostly occurred in the buffer regions of the PA, with entry prohibited in the core zone. We transparently communicated sampling objectives and camera trap placement to local stakeholders, actively involving range/beat-level forest department staff. This approach minimised capturing photos compromising local community individuals' privacy, and we promptly removed any such images from our repository. The data was also shared with the forest

department, requiring no further ethical clearance due to the non-invasive sampling nature.

The study utilised GPS-collar movement data borrowed from the ongoing landscape project, as part of which two tigers were collared in the current study area (permission letter F. No. 1-19/2019 WL, dated 15/09/2020, issued by MoEFCC-Wildlife Division, Government of India). All activities involving tigers' capture, handling, and radio-collaring adhered strictly to animal ethics and protocols approved by the National Tiger Conservation Authority (NTCA). These activities were conducted by qualified and authorised personnel associated with the Wildlife Institute of India and the Madhya Pradesh Forest Department, with minimal adverse effects on the animals.

## Study area

Panna Tiger Reserve (PTR), with an extent between 79°29′ to 80°17′E longitudes and 24°16′ to 24°55′N latitudes, is located in the Biogeographic Province 6A Deccan Peninsula—Central Highlands (*Rodgers, Panwar & Mathur, 2002*), in the central Indian state of Madhya Pradesh (Fig. 1A). PTR is topographically divided into the Ken River valley, where the Ken River cuts through PTR (Mandla and Chandranagar Ranges), the Hinauta plateau (Hinauta Range) and the Talgaon plateau (Panna Range) of the Vindhyan Range. PTR has an area of 1,598 km$^2$, of which 576 km$^2$ is the core (Critical Tiger Habitat) and 1,022 km$^2$ is the buffer zone (Multiple Use Area) (*Madhya Pradesh Forest Department, 2007*, *2012*).

The vegetation type includes Northern tropical dry deciduous mixed forest, Southern tropical dry deciduous teak mixed forest (with the northernmost natural distribution of teak), dry deciduous scrub forest, *Boswellia* forest, *Anogeissus pendula* forest (easternmost natural distribution) and dry bamboo brakes (*Champion & Seth, 1968*). With *Panthera tigris* (Tiger) being the top predator, other co-predators include *Panthera pardus* (Leopard), *Melursus ursinus* (Sloth bear), *Hyaena hyaena* (Striped hyaena), *Cuon alpinus* (Wild dog), *Felis chaus* (Jungle cat) and *Canis aureus* (Golden jackal). The prey species include *Rusa unicolor* (Sambar), *Axis axis* (Chital), *Boselaphus tragocamelus* (Nilgai), *Gazella bennettii* (Indian gazelle), *Tetracerus quadricornis* (Four-horned antelope), *Sus scrofa* (Wild pig), *Semnopithecus entellus* (Northern plains gray langur) among others (*Ramesh et al., 2013*).

The tiger population of PTR was functionally extinct in 2009 due to poaching. In an effort to recover the population, six tigers were translocated from adjacent reserves and reintroduced to PTR between 2009 and 2014 (*Sarkar et al., 2016*), and with adequate supplementation in the initial years, the total population has currently surpassed 60 individuals (*WII, 2022*; *Qureshi et al., 2023*). Between 2005 and 2007, the authorities relocated more than 13 villages from inside the core area of PTR. However, along with tiger numbers, livestock depredation events have also been on the rise (*Kolipaka et al., 2017*). Anthropogenic pressures remain significantly high, as the buffer zone of PTR is home to an estimated human population of over 43,000, along with numerous livestock in 42 villages (*Kolipaka et al., 2015*). The local community has customary access to the buffer forests, which are informally regulated rather than through official government channels. Additionally, there has been a concurrent shift in attitudes toward tigers, transitioning

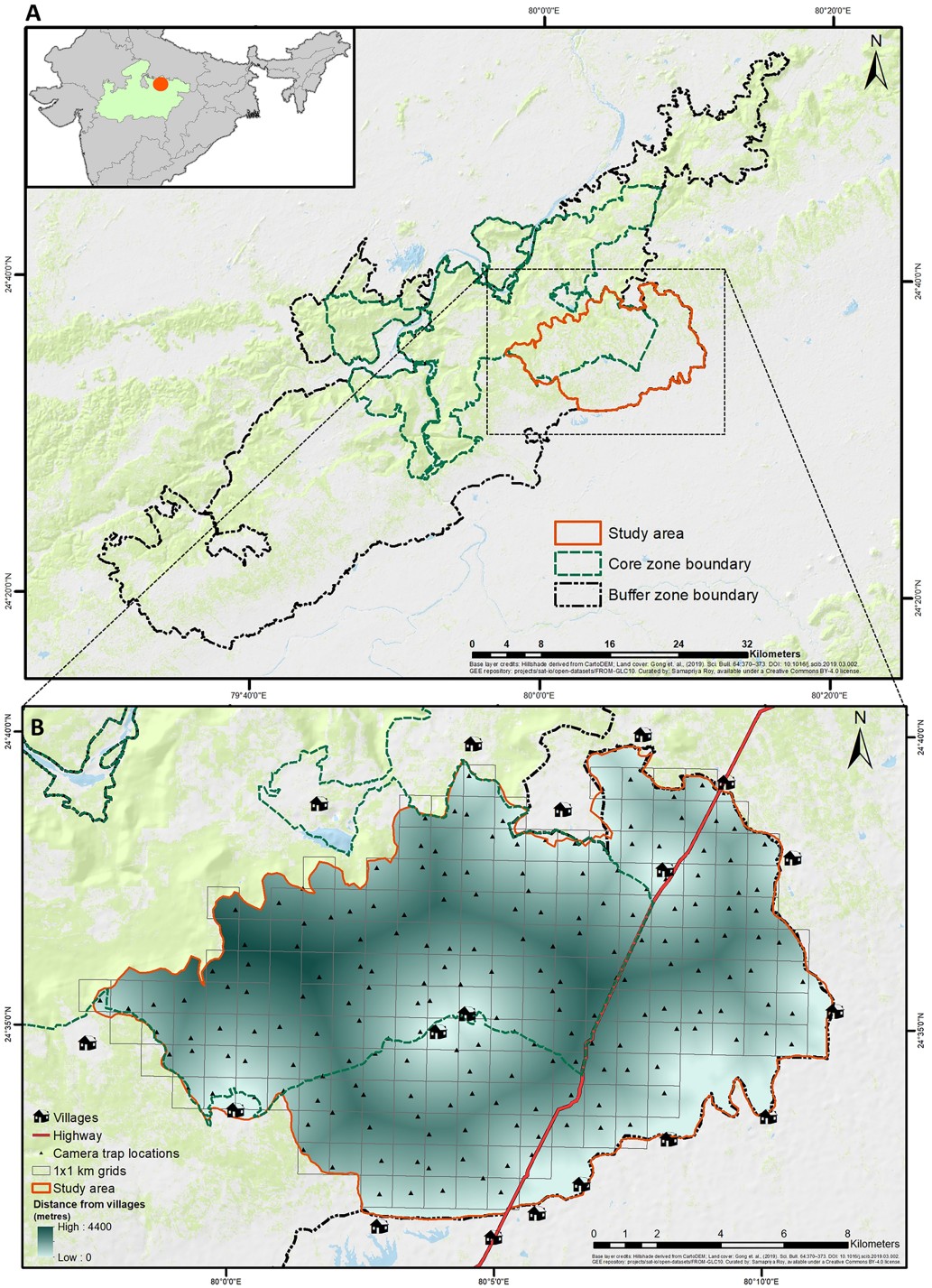

**Figure 1 Geographical location of study area in Panna tiger reserve, Madhya Pradesh, central India.** (A) (top) The larger extent of Panna Tiger Reserve in the central Indian landscape, shows the core, buffer and study area. (B) (bottom) The smaller extent of the study area depicts the sampling design, anthropogenic features and the gradient of distance to surrounding villages. Base layer credits: *NRSC (2021)*; FROM-GLC10.

from a tolerant perspective (*Kolipaka et al., 2015*) to a more negative perception (*Malviya, Kalyanasundaram & Krishnamurthy, 2022*).

The current study was designed and executed in an area of 200 km$^2$ in the southeastern part of PTR, predominantly in the winter of 2020–2021. Surrounded by human settlements (villages) and intersected by State Highway 49 (SH), the area consisted of an equal proportion of core and buffer ranges (Fig. 1) with variable ranges of human disturbance, livestock presence, and tiger movement. The selected area also witnessed about 42% of all livestock depredation events by tigers, according to the compensation records in PTR from 2018 till the initiation of the study (Data S1). *Malviya & Krishnamurthy (2022)* have already identified a significant part of the current study area as a conflict hotspot based on fine-scale risk mapping in PTR.

## Field methods

Technological advancements have given an edge to movement ecological research (*Kays et al., 2015*; *Prakash et al., 2022*), with most of the recent studies utilising resource-intensive GPS-tracked movement data to explain differential behaviour in human-dominated landscapes (*Habib et al., 2021*; *Serieys et al., 2023*). However, in the current study, we present a cost-efficient and non-invasive method to explore our objectives. We employ a fine-scale camera trap survey that can be replicated at the local population-level and use GPS-movement data only to complement our results, such as in showcasing individual variations and differential movement rates near human settlements.

**Camera trap survey:** To understand animals' spatio-temporal movement and space use at as fine a scale as possible, we deployed camera traps intensively at a 1 km × 1 km grid size throughout the study area. The survey spanned from late December 2020 to early April 2021. The cameras were placed evenly on the most suitable animal trails or forest paths, as close to the centroid of the representative grids as possible. They were fixed at a height of 0.5–1 m, depending on the terrain, so that the plane of the photograph was parallel to the trail to capture the flank region of animals and enable individual identification of tigers. We avoided areas within or very close to human settlements to prevent theft and vandalism. Utilising Cuddeback C-1 camera traps with an active white flash, we set each camera to capture photos at 5 to 15 seconds intervals, considering area openness, the risk of false captures, and the intensity of human activity during the day, and Fast-As-Possible mode at night. Cameras underwent checks for vandalism, malfunctions, or battery drainage, with data collected at least once every two weeks. We replaced damaged cameras whenever possible.

**Monitoring GPS-collared tigers:** Two tigers within the study area were collared in the winter of 2020–2021 as part of a larger landscape project (*WII, 2022*). The first, a 27-month-old female tiger (P213-63), and the second, a 14-month-old male tiger (P234-31), were selected for collaring based on their regular movement outside PTR's core areas, often near human settlements. Equipped with GPS/Satellite/VHF (Vetronics Vertex) radio collars, both transmitted location data at 0.5–1 h intervals, enabling detailed movement monitoring and analyses.

**Livestock-kill/GPS-cluster site inspection**: To understand the spatial patterns of livestock depredation, a site inspection was conducted at clusters of GPS fixes of collared tigers and known locations of livestock depredation events during the study period. Clusters for inspection were based on concentrated GPS fixes in an area spanning two or more days, within a radius of 50–100 m, as previous studies in the central Indian landscape have recorded the average drag distance of livestock kills by tigers at 50 ± 54 (S.D) metres (*Miller, Jhala & Jena, 2016*). We inspected the sites only after the tiger had moved away from the suspected kill sites and used all locations with kill remains present for analysis. The kills during the study period, intimated informally by forest guards and villagers, were also readily inspected. Further, we procured the records of livestock kill compensation granted to the owners from the forest department and used all the locations within the study period season since 2018 in the analysis.

## Analytical methods

**Processing camera trap data:** Consecutive captures at intervals of 30 min was considered an independent capture for each species. We calculated the relative abundance index (RAI) of species in consideration as capture rates (CR: independent captures per trap night), scaled to 100 trap nights (*O'Brien, Kinnaird & Wibisono, 2003*). We identified the number of unique individual (NI) tigers that were photo-captured based on the stripe patterns and determined the gender according to the catalogue maintained by the forest department. In the context of the current study, wild prey is referred to as the combination of Sambar, Chital, Wild pig and Nilgai, as they are known to be the preferred prey of tigers with the highest abundances in PTR (*Ramesh et al., 2013*; *Chundawat, 2018*); domestic prey/livestock includes goat, buffalo, cow and ox. We conducted all camera trap data management and processing using MS Excel (*Microsoft Corporation, 2016*) and the R package: camtrapR (*Niedballa et al., 2016*) in R statistical environment (*R Core Team, 2023*) through RStudio desktop IDE (*Posit Team, 2023*).

**Processing spatial data:** We created multiple raster layers to extract different spatial covariates within the study area. We used Sentinel-2 MSI: MultiSpectral Instrument, Level-2A (*ESA, 2021*) with 10 m spatial resolution imagery of late February 2021 as the base satellite imagery for calculating various indices such as NDVI, NDWI, and preparation of Land Use Land Cover (LULC) layer. Water bodies were digitised from the prepared LULC, NDWI layer, supplemented by sources identified during fieldwork. CartoDEM with 30 m spatial resolution was acquired from the Bhuvan portal of National Remote Sensing Centre (*NRSC, 2021*) to extract elevation and slope. To measure distances to anthropogenic features, we digitised village boundaries and roads in the study area from Google Earth Pro (ver. 7.3.3.7786). For creating maps representing a continuum of areas between high and low values, we employed Empirical Bayesian Kriging (EBK) (*Krivoruchko, 2012*) as the spatial interpolation method. All raster calculations and processing were done in Google Earth Engine (*Gorelick et al., 2017*) and ArcGIS 10.6 (*ESRI, 2021*).

**Statistical tests and analyses:** We checked for all assumptions, such as normality and multicollinearity among variables of datasets, before performing each analytical test. We retained variables after evaluating the variance inflation factor (VIF) and ecological

significance when correlated. For non-normal datasets, the statistically significant difference between two sample sets was checked using the Mann-Whitney U test or the Kruskal-Wallis test, depending on the number of sample sets (*Sokal & Rohlf, 1995*). We used a generalised linear model (GLM) framework with a suitable link function and appropriate distribution family assessed by a Goodness-of-Fit test to model the effect on a response variable whenever multiple predictor variables with non-/linear relationships or of non-/normal nature existed. To improve model convergence and account for quadratic and interactive terms, we standardised all continuous predictor variables before conducting analyses. We performed all modelling and model selection using R packages: MuMIn (*Bartoń, 2019*), MASS (*Venables & Ripley, 2002*), model diagnostics using DHARMa (*Hartig, 2021*), and data visualisation using ggplot2 (*Wickham, 2016*), ggbreak (*Xu et al., 2021*) and jtools (*Long, 2023*) in R statistical environment (*R Core Team, 2023*) through RStudio desktop IDE (*Posit Team, 2023*). The best-fit model was selected based on the lowest AICc criteria built by sequentially eliminating uninformative parameters (*Arnold, 2010*). Covariates with significant beta-coefficients were considered the strongest predictors of the response variable. We avoided model averaging among top models (models within 2ΔAICc of the best-fit model) (*Burnham & Anderson, 2002*) whenever covariates in each model were uninformative, based on the zero-value overlap of their 95% confidence interval estimates (*Galipaud et al., 2014*; *Leroux, 2019*).

### Spatio-temporal activity of tigers with respect to human disturbance

The study area experienced substantial anthropogenic activities, including fuelwood and fodder collection within a 2 km radius of village boundaries, and livestock herding extending up to 5 km from the core of villages into PTR, particularly during the winter season (*Kolipaka et al., 2017*). Our camera trap sampling at every 1 km distance enabled us to visualise and explore variations in tiger activity within such fine-scales, akin to the scales adopted by *Carter et al. (2012)*. The adoption of 1 km was also in line with the scale of ESZ delineation around PAs, as required in India (*TH, 2022*; *SCI, 2023*). For this purpose, we graded the study area into distance classes of 1 km width based on distance from villages, resulting in four classes: 0–1, 1–2, 2–3 and 3–4 km, and had 90% of all deployed camera traps within 3 km distance from village boundaries (Table S1). For all modelling purposes and to draw definitive inferences regarding the effect of human disturbance on tiger activity at the population-level, we treated distance to villages as a continuous predictor rather than binning into distance classes.

*Proportion of area occupied, probability and intensity of space use*
We first determined the proportion of area used by tigers by modelling the effect of habitat covariates on the probability of tiger space use in an occupancy framework (*MacKenzie et al., 2002*) in the Program PRESENCE (version 2.13.11) (*Hines, 2006*). The occupancy model estimates two variables–$\psi$ (psi), the proportion of area used or the probability of species occupancy at a site, and $\rho$ (phi), the probability of detecting a species if it is present. We used camera trap data from the first 100 days of sampling to build detection histories of tigers and considered each day as one sampling occasion, resulting in 100 occasions. We

used remotely sensed variables at 100 m scale and RAI of each wild prey species as separate variables to model the probability of space use. To model detection probability, we used the finest scale resolutions available for site-specific covariates such as NDVI (10 m), elevation and slope (30 m). In addition, we used camera trap effort-based (number of sampling days) and location-based (binary; animal trails coded as 1 and forest paths as 0) variables to model detection probability. We used R package: unmarked (*Fiske & Chandler, 2011*) to generate confidence intervals and plot the marginal effects of covariates.

Subsequently, we aggregated the camera trap data within each 1 km distance class to explore the variation in the intensity of space use of tigers with increasing distances from villages through box-whisker plots and statistical tests to determine group differences. We assessed the variation in NI, RAI, and the proportion of male and female tiger RAI across distance classes.

*Modelling temporal activity and overlap analysis*

We tested for statistically significant effect of disturbance factors (distance to villages and highway; livestock and human RAIs) on tiger activity time, using Bayesian circular GLM (*Mulder & Klugkist, 2017*). The model uses an arctangent link function to predict a circular response, and we determined the best model based on the lowest Watanabe-Akaike information criterion (WAIC) value. We defined the activity time as being more or less nocturnal based on the hours-to-noon measure (*Nickel et al., 2020*). All temporal pattern analyses were conducted after testing for any seasonal shifts in tiger activity time during the study period (between January-February and March-April) using Fisher's non-parametric test for common median direction. We performed all circular statistical modelling and tests using the R packages: circglmbayes (*Mulder & Klugkist, 2017*), circular (*Agostinelli & Lund, 2013*) and methods and functions described by *Pewsey, Neuhäuser & Ruxton (2013)*.

Further, we conducted an activity overlap analysis of humans, livestock and wild prey with tiger temporal activities at each distance class to visualise how tiger activity changes with respect to others. We estimated the diel activity patterns using a non-parametric kernel density method (*Worton, 1989*) and used the kernel density-based coefficient of overlapping, $\Delta$, to measure the degree of overlap. We used $\hat{\Delta}_1$ as the estimator when the smaller sample was <75 and $\hat{\Delta}_4$ when it was >75 (*Schmid & Schmidt, 2006*; *Ridout & Linkie, 2009*). We performed all temporal pattern analyses and activity overlap plotting using R packages: camtrapR (*Niedballa et al., 2016*) and overlap (*Ridout & Linkie, 2009*).

## Movement range and patterns using GPS telemetry

We acquired all the GPS locations of collared tigers within the study period (January to April 2021) from Vectronic Aerospace (*Inventa Wildlife Monitoring, 2021*). We resampled all GPS fixes to hourly intervals and excluded irregular fixes from the analysis. To understand the extent of tigers' movement near human settlements, we first performed a home range analysis to deduce overlap of their range with villages. We used autocorrelated kernel density estimation (AKDE) to calculate home ranges. Subsequently, we used distinct distance classes (0–100, 100–250, 250–500, and >500 m from villages),

following the comparison by *Athreya et al. (2014)*, to examine the proportion of GPS locations inside or near villages. Further, we segregated the GPS locations into dawn, day, dusk and night periods based on sunrise, sunset and astronomical twilight times using R package: suncalc (*Thieurmel & Elmarhraoui, 2022*). During each period, we measured the animals' distance to human settlements and displacement (step length per unit time) in metre/hour (m/h) and analysed the variation. Given the skewed, non-normal nature of displacement, we used median step lengths (MSL) and median absolute deviance (MAD) to gauge data spread. We compared the total number of GPS locations within 250 and >250 m from villages to identify differences in movement rates near and away from human settlements. We conducted all home range and GPS movement pattern analyses using the R packages: ctmm (*Fleming & Calabrese, 2023*) and amt (*Signer, Fieberg & Avgar, 2019*).

### Modelling depredation risk and assessing predator's influence

The proportion of livestock depredation by tigers is often higher in winter (*Malviya & Ramesh, 2015*; *Borah et al., 2018*). Hence, we collected information on livestock kill sites during the study period and season, including those from the forest department compensation records. The selection of the current study season also enabled us to understand the influence of tiger presence on depredation events. We converted the livestock kills sites' information into presence-absence format, in which each 1 km × 1 km grid-cell with at least one depredation event was given a value of 1 and those without an occurrence as 0. Logistic regression models (binomial GLM) were then run with site-specific covariates based on field observations and previous studies that indicated an association with depredation events (*Soh et al., 2014*; *Miller et al., 2015*; *Miller, Jhala & Jena, 2016*; *Rostro-García et al., 2016*; *Malviya & Krishnamurthy, 2022*). We measured variables such as distance to anthropogenic features and water; NDVI, slope and elevation within 100 m buffer from kill locations for kill-present grid-cells and camera trap locations for kill-absent grid-cells. We based the 100 m buffer width on the known kill drag distance of tigers in the landscape (*Miller, Jhala & Jena, 2016*) and considering the scale at which ambush predators such as tigers operate during a kill (*Miller et al., 2015*). We also explored interactive and quadratic relationships of different variables since variables such as distance to roads and villages are known to have a threshold relationship with livestock depredation incidents (*Miller, Jhala & Jena, 2016*).

To ascertain the predator's role in the conflict, we used tigers' space use probability and relative abundance within the grid as predictor variables. However, since tigers' space use probability was a derived parameter from many of the same covariates, we used AICc+VIF criterion to identify the optimal model, effectively addressing multicollinearity. This involved initially ranking models based on AICc and subsequently removing any models with VIF values exceeding five as top model candidates. This approach was also suitable since our primary focus was to understand the effect of covariates rather than solely achieving an accurate prediction of livestock depredation probability. We evaluated the best-fit model by plotting the area under the curve (AUC) and calculated the receiver operating characteristic (ROC) value using the R package: Deducer (*Fellows, 2012*). AUC

value corresponds to an integral of the accuracy of the logistic model at every possible threshold value between 0 and 1.

**Conflict-risk mapping:** We calculated the probability of livestock depredation at each grid-cell by back-transforming the predicted probability values and interpolating them to the entire study area to represent them spatially. The transformation of a logistic link is given by the formula: $y = e^x/(1 + e^x)$.

# RESULTS

We deployed 191 camera traps across the study area for a total effort of 15,607 trap nights, resulting in 53,484 independent captures. A total of 878 independent photo-captures of tigers were recorded, that included 17 unique individuals (Table S1). In addition, at least 29 wild mammal species were photo-captured during the entire sampling period (Table S2). The average number of camera trap sampling days was 81.7 ± 12.7 (S.D).

## Occupancy patterns and spatial variability along a disturbance gradient

Occupancy analysis revealed 76% of the study area being used by tigers (naïve $\psi$ = 0.733; covariate/detection-corrected $\psi$ = 0.756; Bayesian 95% CI [0.733–0.921]). According to the best-fit model, the distance to villages and RAI of sambar positively affected the probability of space use by tigers (Table 1; Fig. 2), whereas detection probability increased with effort, higher distance to villages, proximity to highway, higher vegetation cover (NDVI), and placement of camera trap on forest paths (Table 1; Fig. S1).

The boxplots showed a positive trend in RAI as well as the number of unique individuals captured with increasing distance classes from villages (Fig. S2), with statistically significant variation across distance classes (RAI: H = 32.34, df = 3, $P < 0.001$; NI: H = 25.8, df = 3, $P < 0.001$). There was also significant variation in the proportion of male and female tiger RAI across distance classes ($\chi^2$ = 9.35, df = 3, $P$ = 0.025), with the proportion of female captures decreasing with distance to villages.

## Temporal activity: disturbance effects and overlap trends

We ruled out any seasonal variation within the study period due to non-significant test results (Fisher's non-parametric test: Pg = 0.626, df = 1, $\chi_c^2$ = 3.84, $P$ = 0.428) (see Table S4 for details of the test). Circular GLMs revealed that, among all the disturbance factors, only distance to villages exerted a statistically significant effect on tiger activity time. According to the best-fit model, when the predictor value is at zero, the slope of the counter-clockwise change of tiger activity time is −0.114 (Bayesian 95% CI [−0.177 to −0.051], generated with 10,000 Markov chain Monte Carlo iterations) (Table S5). The circular intercept value was measured at 3.03 radians, which indicates the origin of the activity shift ($\mu$). Represented in terms of the time of day, the mean tiger activity occurred at 23:34 h at mean distance from villages. The model-predicted activity time of tigers showed a clockwise shift (towards and beyond midnight hours) at lower distances and an anti-clockwise shift (towards early night hours) at higher distances from the mean activity time and distance (Fig. 3).

**Table 1 Occupancy models considered to determine tigers' probability of space use and probability of detection.**

| Models | Covariates used for modelling Ψ | | | | | Covariates used for modelling ρ | | | | | | | Model information | | | |
|---|---|---|---|---|---|---|---|---|---|---|---|---|---|---|---|---|
| | Intercept | Distance to villages | RAI | | NDVI | Intercept | Effort | Distance to | | NDVI | RAI Humans | Location | df | AICc | ΔAICc | LL |
| | | | Sambar | Humans | | | | Villages | SH | | | | | | | |
| Best-fit | 1.340 | **0.627** | 0.716 | – | – | **−2.681** | 0.076 | **0.378** | **−0.255** | **0.094** | – | **−0.300** | 9 | 5,968.90 | 0.00 | −2,974.93 |
| Within 2ΔAIC | 1.317 | **0.626** | **0.697** | – | – | **−2.652** | – | **0.384** | **−0.233** | **0.095** | – | **−0.336** | 8 | 5,969.30 | 0.40 | −2,976.25 |
| | 1.338 | **0.627** | **0.611** | – | 0.156 | **−2.681** | 0.076 | **0.378** | **−0.255** | **0.093** | – | **−0.298** | 10 | 5,970.70 | 1.80 | −2,974.72 |
| | 1.418 | **0.682** | **0.799** | 0.509 | – | **−2.686** | 0.080 | **0.384** | **−0.256** | **0.094** | 0.008 | **−0.303** | 11 | 5,970.80 | 1.90 | −2,973.66 |
| Null | Ψ(.) | | | | | ρ(.) | | | | | | | 2 | – | – | – |
| Global | Ψ (**Distance to villages** + Distance to SH + Distance to water + RAI Livestock + RAI Humans + **RAI Sambar** + RAI Chital + RAI Nilgai + RAI Wild pig + Elevation + NDVI + Slope), ρ (Effort + **Distance to villages** + **Distance to SH** + Elevation + NDVI + Slope + RAI Humans + **Location**) | | | | | | | | | | | | 22 | 5,990.74 | 21.64 | −2,970.36 |

Note:
The top row indicates covariates in the top models and model information. Each subsequent row indicates different models: from the best-fit model at the top to the global model at the bottom. The cell value under each variable is its untransformed parameter estimate when it was present in the model. Values in bold indicate that their 95% confidence intervals do not overlap zero (Table S3).

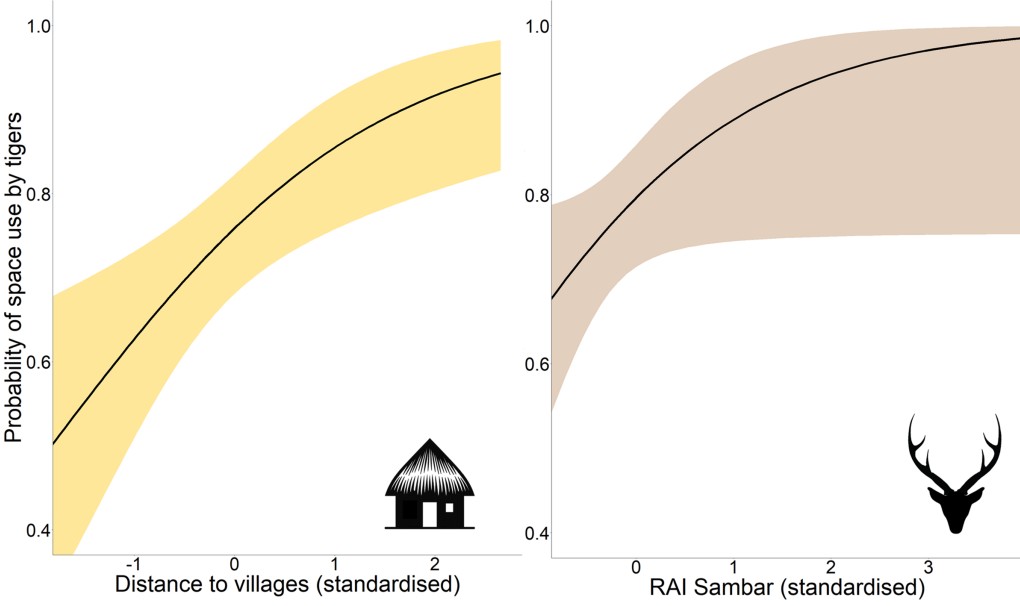

**Figure 2 Marginal effect response curves of predictors influencing tigers' space use probability.** The plotted curves represent predictors identified by the best-fit occupancy model. The shaded region indicates the 95% confidence interval of the response curve. Identical Y-axis limits are maintained for direct comparison between predictor responses. Response curves of detection probability predictors are given in Fig. S1. Graphics/silhouettes in the figure generated with the assistance of DALL·E 2.

Activity overlap analysis showed a decreasing trend in human-tiger overlap as distance to villages increased (Fig. S3). The coefficient of overlap decreased from 0.381 to 0.304. Livestock-tiger activity overlap was highest at the closest distance class of 0–1 km at 29%, which decreased to 15.6% and 17.4% at 1–2 and 2–3 km distance classes, respectively.

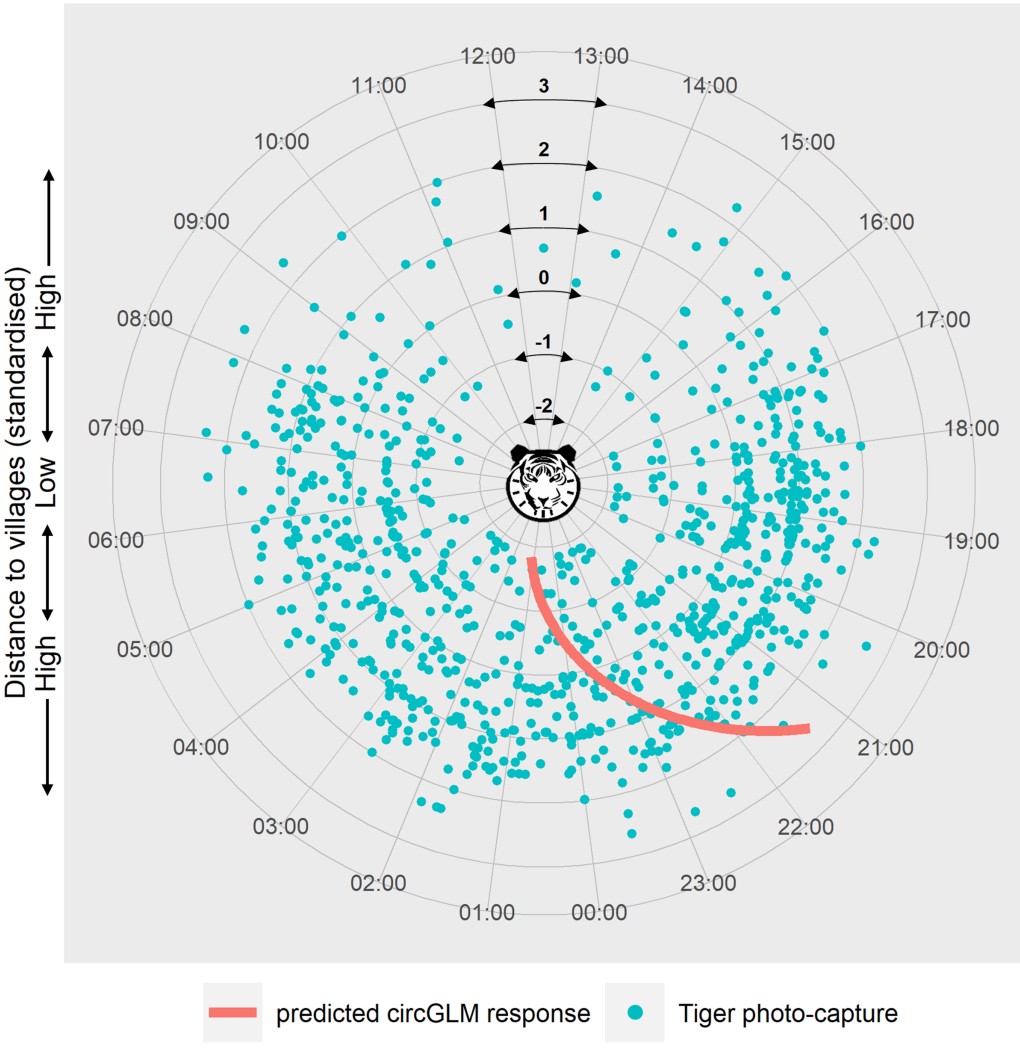

**Figure 3 Circular plot of modelled temporal activity of tigers.** Each dot represents an independent photo-capture of a tiger against distance to villages, with the capture time labelled around the circumference. The concentric circles at successive radii indicate the standardised distances to villages, with the lowest distance at the centre and the highest towards the periphery of the plot. The curve depicts the model's predicted temporal response of tiger activity against distance to villages. The response curve was fitted using an arctangent link function within a Bayesian circular GLM framework. The slope of the response ($-0.114$) was calculated at the midpoint of the link function, corresponding to the zero value of the predictor (where $\mu = 3.03$ was calculated). The modelled response curve shows tiger activity around 09:15 pm, at the highest distance from villages that shifts clockwise past 01:00 am in proximity to villages. Graphic/silhouette in the figure generated with the assistance of DALL·E 2.

Notably, the wild prey-tiger activity overlap also showed a negative trend, decreasing from a high 86% at 0–1 to 56% at 3–4 km.

## Differential movement patterns near human settlements

A total of 8,110 GPS locations from January–April 2021 of both tigers were available for analysis. AKDE home range estimates of the two collared tigers showed that, for juvenile male, 10% (2.86 km$^2$) of the total home range area (28.13 km$^2$; 95% CI [17.00–42.03]) was

within villages, whereas, for the female, around 12% (4.86 km$^2$) of the total home range area (41.42 km$^2$; 95% CI [35.49–47.80]) was within villages. A considerable proportion of GPS locations of the tigers were inside or very near villages. (8 + 10)% of all the GPS fixes were within 0–100 m, and (7 + 3)% of the locations were within 100–250 m of villages for the juvenile male and female tigers (Fig. S4).

Distances to villages at dawn, day, dusk and night periods were comparable for the juvenile male tiger (Table S6). However, the female tiger was closer to the villages in the dark hours compared to the dawn and day periods, with at least 77% decrease in distance to villages at night compared to day (Table S6). Displacement of both tigers in day and night differed significantly (male: U = 587,722, $P < 0.001$, MSL 1.7 times higher in day; female: U = 829,883, $P < 0.001$, MSL 10 times higher at night) (Table S7). Both tigers displayed significantly higher displacement near villages (<250 m) compared to higher distances (>250 m) (male: U = 515,246, $P < 0.001$, MSL-25 m/h *vs* 15 m/h; female: U = 445,766, $P < 0.001$, MSL-154 m/h *vs* 27 m/h) (Fig. S4).

## Conflict probability prediction and risk mapping

A total of 252 livestock depredation events by tigers were recorded in PTR from 2018 till October 2020, until which forest department compensation records were available (Data S1). Of these, 106 were within the study area, and 45 were within the study season. Additionally, we recorded eight sites from cluster site and direct depredation event site inspections. Finally, 47 kill locations laid within 1 km × 1 km sampling grids resulted in a total of 34 presence-grids (Data S2).

The best-fit model for predicting depredation probability revealed that distances to villages ($P = 0.002$) and highway ($P = 0.003$) negatively affected the probability, with distance to villages also exhibiting a positive quadratic effect ($P = 0.039$) (Table 2; Fig. S5). Tigers' space use probability had a positive influence ($P = 0.056$, but, its 95% confidence interval limits did not overlap zero), while the RAI of wild prey, though present in the best-fit model, showed a non-significant effect ($P = 0.134$) (Table 2; Table S8). The hotspots of depredation risk were predominantly in areas which were close to both villages and highway (Fig. 4). The AUC value for the best-fit model was estimated at 0.795, indicating fairly high discriminatory power of the model (Fig. S6), *i.e.*, the model correctly classifies a randomly drawn pair, each from a depredation-present and depredation-absent 1 km × 1 km grid-cell groups ~80% of the time.

## DISCUSSION

The current study establishes and reinforces the increasing impacts of anthropogenic disturbance on carnivores, inducing behavioural shifts and vulnerability. Our findings offer compelling evidence of spatio-temporal activity adjustments in response to human influences, supporting our hypotheses. We identify the distance to human settlements as the most influential variable through spatial, temporal, movement, and conflict-risk analyses. Tigers exhibited a higher probability of space use at greater distances from villages, facilitated by higher relative abundance of sambar. Tigers were also more nocturnal in proximity to villages. Our results show that tigers avoid human influences at

**Table 2 Binomial generalised linear models considered for predicting the probability of livestock depredation by tigers.**

| Models | Intercept | Distance to | | | | RAI | | | | Psi tiger | Model information | | | | |
|---|---|---|---|---|---|---|---|---|---|---|---|---|---|---|---|
| | | Villages | Villages² | SH | Water | Humans | Livestock | Wild prey | Tiger | | VIF | df | AICc | ΔAICc | LL |
| Best-fit | **−2.548** | −1.257 | **0.486** | −0.912 | – | – | – | −0.752 | – | **0.777** | 3.54 | 6 | 156.28 | 0.00 | −71.91 |
| Within 2ΔAICc | −2.006 | −0.817 | – | −0.912 | – | – | – | – | – | – | 1.00 | 3 | 156.78 | 0.49 | −75.32 |
| | −2.078 | −0.906 | – | −0.934 | −0.401 | – | −0.239 | – | – | – | 1.17 | 5 | 157.13 | 0.85 | −73.40 |
| | −2.343 | −0.945 | 0.372 | −0.989 | – | – | – | – | – | 0.360 | 2.29 | 5 | 157.26 | 0.97 | −73.47 |
| | −2.034 | −0.929 | – | −0.907 | – | – | – | – | 0.264 | – | 1.23 | 4 | 157.61 | 1.32 | −74.70 |
| | −2.555 | −1.218 | **0.489** | −0.940 | – | – | −0.175 | −0.736 | – | 0.728 | 3.63 | 7 | 157.80 | 1.51 | −71.59 |
| | −2.587 | −1.197 | 0.477 | **−0.917** | −0.334 | −0.142 | – | −0.749 | – | **0.788** | 3.62 | 8 | 158.15 | 1.87 | −70.68 |
| Null | −1.530 | – | | | | | | | | – | | 1 | 180.94 | 24.65 | −89.46 |
| Global | Depredation probability~Elevation + NDVI + Slope + **Distance to villages** * RAI Livestock + **Distance to SH** + Distance to water * RAI Livestock + RAI Humans * RAI Livestock + RAI Wildprey * RAI Tiger + psi Tiger + I (Distance to villages^2) + I (Distance to SH^2) | | | | | | | | | | 5.61 | 18 | 174.59 | 18.31 | −67.31 |

Note:
Table interpretation remains same as that of Table 1. Values in bold indicate that their 95% confidence intervals do not overlap zero (Table S8).

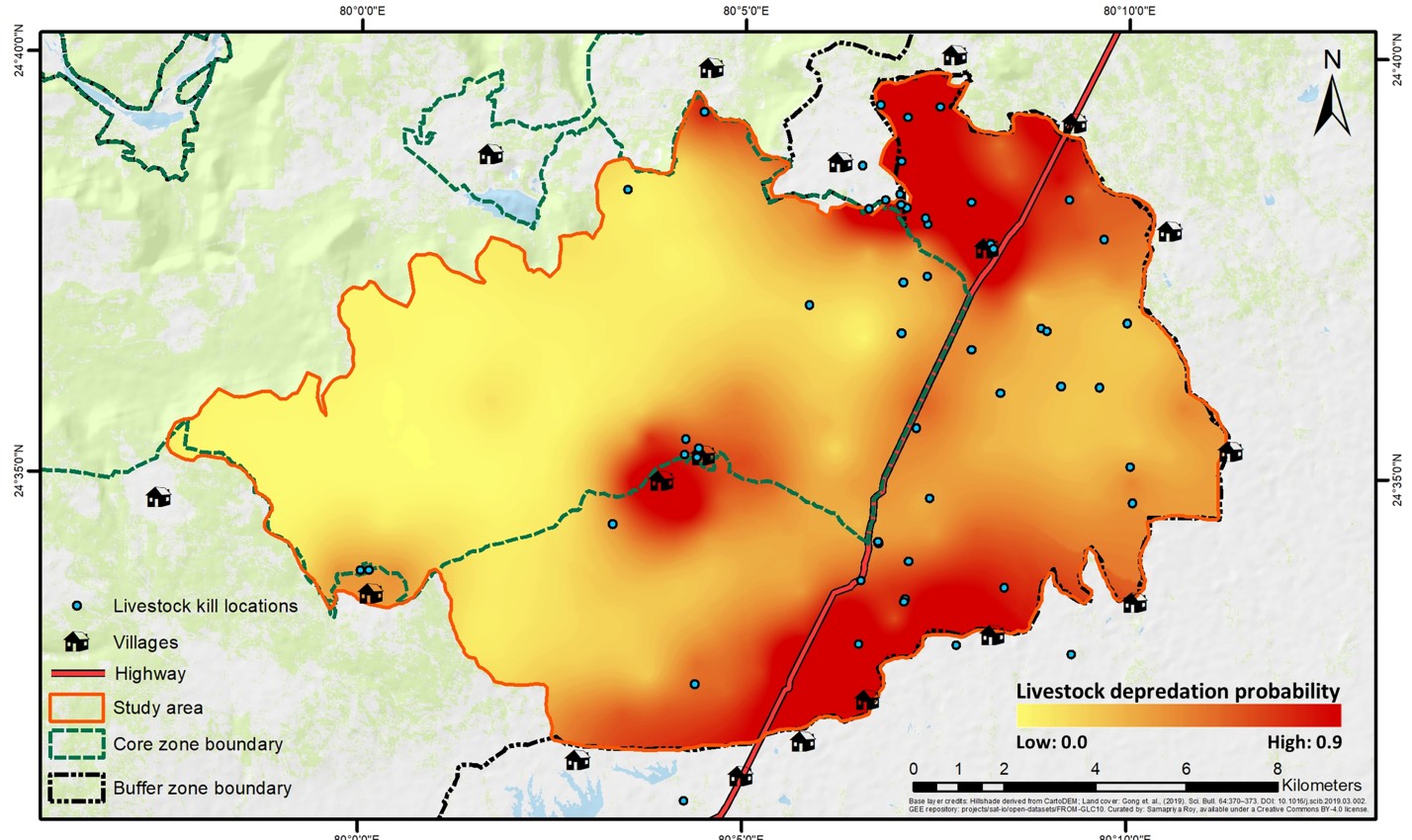

**Figure 4 Spatial representation of livestock depredation risk by tigers.** The continuous surface raster has been created by spatially interpolating the back-transformed values of depredation probability from the best-fit predation-risk model. Base layer credits: *NRSC (2021)*; FROM-GLC10.

the local population-level. However, with PTR recording historically high tiger numbers, a larger proportion of sub-adult tigers now need to navigate the human-dominated areas for exploration and dispersal, where the presence of free-ranging domestic prey is significant. While the movement pattern analyses are constrained by the data on only two individual tigers, it is evident that they display exploratory use of areas near villages, with approximately 11% of their home range falling within village boundaries. Such behaviour potentially increases the depredation risk of domestic prey, as indicated by the contributing effect of tigers' space use probability in predicting conflict-risk. Our study distinguishes the cause-effect relationships of tiger activity shifts and conflicts, indicating that, while tigers predominantly exhibit avoidance behaviour toward human disturbance, the occasional encounters with livestock present opportunities for depredation. The fine-scale variability in behavioural and conflict patterns revealed by our study offers crucial insights for land management and conflict mitigation.

## Tiger space use shifts in response to disturbance

Despite substantial disturbance, a relatively high proportion (76%) of the total 200 km$^2$ study area was used by tigers, with the upper limit as high as 92%, revealing high potential occupancy. Due to our systematic fine-scale sampling, the detection-corrected occupancy varied only by 2%. The relative abundance of a large-bodied wild prey–sambar, and distance to villages positively affected tiger occurrence. *Mills et al. (2020)* have observed a large carnivore's space use being primarily dependent on prey availability, even in the presence of human activity. Our results agree with the observation while also arguing that the ability to effectively utilise the habitat for prey is limited by anthropogenic pressures–a conclusion supported by other studies on tigers (*Chanchani et al., 2016*; *Wang et al., 2018*) and other large carnivores (*Oriol-Cotterill et al., 2015a*; *Gehr et al., 2017*). Anthropogenic disturbance can also indirectly affect tiger occupancy by negatively influencing the activity and distribution of their primary prey species (*Soh et al., 2014*; *Xiao et al., 2018*, but see *Berger (2007)* and *Lamichhane et al. (2023)* for evidence of "human-shield effect" on prey). As a result, the carnivores might adjust or even enlarge their home ranges (*Carter, Levin & Grimm, 2019*), further increasing their encounters with humans. *Bhattacharjee et al. (2015)* have shown evidence of higher stress levels in tigers due to human disturbance factors. Previous studies on tigers have also demonstrated a pattern of avoidance of human settlements and infrastructure (*Sarkar et al., 2017*; *Carter et al., 2023*). Inviolate spaces are imperative for tiger populations to thrive (*Harihar et al., 2020*). However, with the irrepressible sprawl of human settlements, habitat amelioration and effective management also take precedence for conserving tigers in PAs (*Linnell, Swenson & Anderson, 2001*; *Carter et al., 2012*). Evidence shows that refugia play a vital role in offsetting the restricted movement and increased vulnerability of large carnivores caused by human pressures (*Oeser et al., 2023*).

Tiger abundance indicated by RAI showed a positive trend across increasing distance classes from villages (Fig. S2). The pattern of a lower abundance of tigers at the highest anthropogenic disturbance has been observed globally (*Johnson et al., 2006*; *O'Brien, Kinnaird & Wibisono, 2003*) and within the country (*Karanth et al., 2011*). While we

acknowledge the limitation of RAI as it does not account for detection, the marginal difference in detection-corrected occupancy metrics in our study improves the credibility of RAI results. Further supporting this claim, our findings indicate that the elevated RAI at increasing distances from villages is not merely a result of a limited number of individual tigers intensively using less-disturbed areas. Rather, more tigers were using areas at higher distances from villages (Fig. S2). As expected, we observed a higher proportion of female photo-captures in all distance classes due to our fine-scale sampling and smaller home ranges of females. However, the distance class-wise analysis revealed an unusual pattern– the proportion of female photo-captures was highest in the nearest distance class to villages compared to the farthest (Fig. S2). Female tigers show higher site fidelity, and female progenies exhibit parapatric nature of home-range, and are primarily dependent on the resources available in an area, requiring secure sites for raising their cubs (*Sunquist, 1981*; *Smith, McDougal & Sunquist, 1987*), as has also been the case in PTR (*Kolipaka et al., 2017*; *Sarkar et al., 2017*; *Chundawat, 2018*). Research has also revealed the higher stress response of female tigers in areas with anthropogenic disturbance (*Bhattacharjee et al., 2015*). In a Tropical Dry Deciduous habitat like PTR, resources such as water availability are scarce and seasonal. The higher proportional abundance of female tigers near villages in the current study may indicate a more complex interplay of resource availability and reproductive strategies, including more female home ranges and disturbance at finer scales than previously understood. Low deterrence by human habitations or high human density on females has also been observed in other large-carnivores (*Wilmers et al., 2013*; *Alegre et al., 2023*).

## Temporal activity influenced by nature of disturbance

In forested habitats that remain undisturbed, predator activity is mainly influenced by that of their prey (*Karanth & Sunquist, 2000*; *Ramesh et al., 2012*). Similar patterns have been observed in tiger habitats affected by human activity (*Yang et al., 2019*). However, our findings indicate that, at fine-scales, there are deviations from this pattern in human-dominated areas, and the nature and impact of disturbances play a more significant role. Our analysis revealed that, among the various disturbance factors, such as distance to villages and highways and the relative abundances of humans and livestock, only the distance to villages had a statistically significant effect on tiger activity time. Model predictions revealed that tigers tend to be more nocturnal at closer distances to human settlements, agreeing with global patterns of higher nocturnality of large carnivores due to predominantly diurnal human disturbance (*Gaynor et al., 2018*; *Nickel et al., 2020*).

Our use of Bayesian circular GLMs, initially applied in experimental psychology (*Mulder & Klugkist, 2017*), highlights its unique potential in wildlife ecology. The model effectively handles the influence of multiple (linear/categorical) predictors on an animal's circular response (activity time or movement direction), which is often challenging in a frequentist setting. We did not find any significant relationship between human presence (RAI) and the activity time of tigers, substantiating the unrelated effects of human footprint (anthropogenic structures) and human presence on the animals' temporal activity (*Nickel et al., 2020*; *Farmer et al., 2022*).

Activity overlap plots mirrored the lack of significance of human activity, as we observed minimal variation in tiger-human overlap across distance classes. This may suggest the tigers' habituation to low-risk human exposures. Nevertheless, tiger activity overlaps with humans, livestock, and even wild prey decreased with distance to villages, although, the extent of overlap with wild prey remained highest in all classes, as is commonly observed. While circular models have predicted increased nocturnality of tigers at the population-level, a crepuscular pattern also emerges at finer scales near human settlements (Fig. S3). Such a variation may be due to certain disturbance elements still at play, such as night safaris conducted in the buffer regions, which may deter tiger activity at night. This crepuscular activity could also allow tigers to exploit available wild prey while avoiding disturbance, contributing to the observed pattern of highest tiger-wild prey overlap near villages. It is crucial to restrict vehicular movement, particularly at night, to ensure the tigers' successful movement and dispersal, given their predominantly nocturnal nature (Carter et al., 2012).

## Movement patterns to negotiate human-dominated areas

The higher movement rate exhibited by the (female) tiger near human settlements agrees with the movement patterns observed in tigers and other large carnivores within the central Indian landscape (Athreya et al., 2014; Habib et al., 2021) as well as across the world (Kertson et al., 2011; Valeix et al., 2012; Gehr et al., 2017). The tiger also chose to explore the areas in proximity to human settlements under the cover of night, indicated by the reduced distance to villages at night (Table S6), complementing the results of temporal activity analysis. Overall, the animal exhibited its highest displacement in proximity to villages (Fig. S4; Table S7). Faster movement rates exhibited by carnivores near human settlements is a trade-off between risk avoidance and optimal foraging (Donatelli, Mastrantonio & Ciucci, 2022). However, the results for the other tiger in this study differed; its displacement was higher during the day than at night. As a juvenile male tiger, it would likely avoid confrontations with resident dominant males and explore the area during the day. Life history stages and personality differences play a significant role in explaining the variation in behavioural responses to similar stimuli (Wolf & Weissing, 2012). Previous reports from PTR have also highlighted similar differences attributed to the age group and sex of the tigers (Kolipaka et al., 2018). Moreover, the ability of large carnivores to persist in human-modified areas is accompanied by their behavioural plasticity, allowing them to selectively choose areas with reduced risk while still capitalising on available resources (Gehr et al., 2017; Evans et al., 2019). About 16% of the total GPS locations of tigers in the study area were within 100 m of villages, highly contrasting Athreya et al.'s (2014) study results, where only 0.2% of locations were found within the same range for a radio-collared tigress in a human-dominated landscape. These results affirm tigers' persistence through behavioural adaptations and align with patterns observed by Kolipaka et al. (2018) in PTR, indicating a higher likelihood of dispersing or sub-adult tigers using areas near villages.

## Reconciling with livestock depredation

Our analysis identified distances to human infrastructures as the most significant predictors of livestock depredation probability by tigers (Tables 2, S8). The finding aligns with a previous study conducted in PTR, which reported higher livestock depredation occurrences at greater distances from the core-zone boundary into the buffer zone (*Kolipaka et al., 2017*). Our results supplement the predictors of depredation probability identified by *Malviya & Krishnamurthy (2022)* in PTR. While our study has limitations due to the low sample size and the absence of a multi-scale approach, our focus on the conflict hotspot in the interface area reveals a distinct pattern. The condition in PTR has also changed significantly between the two study periods (2011–2015 and 2021). During this time, tiger numbers have continued to rise at an average annual growth rate of ~31% (*WII, 2022*), and there is now negligible feral cattle presence within the core zone, with domestic livestock restricted to areas near villages (Fig. S3). Livestock depredation has since been majorly reported from the buffer zones of PTR (*WII, 2022*; Data S1).

Another study within the central Indian landscape has also shown a decrease in depredation probability with increasing distances to roads and villages beyond a threshold distance of approximately 1 km (*Miller, Jhala & Jena, 2016*). In the current study, the highway intersects the study area for 15 km from the north-eastern to the south-western boundary (Fig. 1B). The linear intrusion allows livestock and their herders easy access to PTR's multi-use buffer zone, besides the human trails emanating from villages. Higher proximity to roads has also been shown to be a key factor in predicting an increased risk of human-wildlife conflict in general within the Indian subcontinent (*Sharma et al., 2020*).

However, a deviation from such a pattern, in which depredation risk increased with increasing distance to villages, has been observed globally in tigers (*Soh et al., 2014*) as well as other carnivores (*Soto-Shoender & Giuliano, 2011*; *Wang et al., 2022*). These opposing global patterns underscore the higher intensity of interaction between wildlife habitats and humans (*DeFries, Karanth & Pareeth, 2010*) and high human infrastructure density (*Miller, 2015*). High spatial overlap of humans and tigers gives rise to 'diffuse' or 'soft' edges in tiger habitats. Such areas are more prone to conflicts than hard edges (*Nyhus & Tilson, 2004*), which worsens with the interface expansion between human habitations and PAs (*Dhungana et al., 2018*).

In addition, our results also reveal the contribution of tiger space use probability on depredation risk (Tables 2, S8). Our spatial analyses have already shown the general avoidance behaviour of anthropogenic features and preference for wild prey by tigers. However, with a potential increase in tiger space use, such as during exploratory use of areas near villages, they may occasionally encounter livestock, elevating the probability of depredation. Notably, the relative abundance or the intensity of tiger occurrence (RAI), did not affect depredation probability, supporting our assumption that chance encounters lead to depredation events. The practice of corralling livestock is not widespread in the villages surrounding PTR. Yet, it is deemed an essential step in mitigating conflicts with tigers, as also recommended by *Kolipaka et al. (2017)*. While financial incentives and compensation

for livestock losses are helpful as a *post hoc* mitigation measure (*Dickman, Macdonald & Macdonald, 2011*; *van Eeden et al., 2018*), they are not without significant administerial flaws and inaccessibility (*Karanth et al., 2012*; *Ravenelle & Nyhus, 2017*). Supplementing incentive approaches with preventive husbandry practices is essential to reduce existing risks. Interventions like fencing and corralling have proven to be more effective and enduring (*Khorozyan & Waltert, 2019*) than deterrents (*e.g.*, light/sound/fladry devices) in areas with high human disturbance (*Blumstein, 2016*).

Although our results do not provide conclusive evidence of low wild prey availability leading to higher depredation probability ($P = 0.134$), previous studies on tigers and other large carnivores confirm the effect (*Bing, Endi & Zhongbao, 2009*; *Bhattarai & Fischer, 2014*; *Khorozyan et al., 2015*), aligning with earlier results from PTR (*Malviya & Krishnamurthy, 2022*). Tigers have been observed to prefer wild prey, even in low densities, compared to livestock (*Reddy, Srinivasulu & Rao, 2004*), previously noted in PTR as well (*Chundawat, Gogate & Johnsingh, 1999*). Hence, we argue that an increase in wild prey abundance through habitat amelioration and concurrent reduction in livestock availability would decrease incidences of livestock depredation, a recommendation similar to *Goodrich (2010)*.

Nevertheless, even with improved management, problem individuals may remain. Such conflict-causing individuals are usually transient or physically weak tigers (*Goodrich et al., 2011*), often represented by a very low proportion of the local population (*Lamichhane et al., 2017*; *Chatterjee et al., 2022*). Therefore, it is crucial to identify, monitor and manage such individuals (*Nyhus & Tilson, 2004*). Identifying such tigers based on their specific social network position is an emerging tool, with data collection readily feasible through camera traps (*Carter, Wilson & Gurung, 2023*). These interventions would prevent them from becoming a more significant issue for the entire species (*Tilson & Nyhus, 1998*). However, in a multi-use landscape, occasional depredation should be expected as a natural event in any prey-predator system that may not involve a problem animal (*Linnell et al., 1999*).

## CONCLUSIONS

Conservation of vulnerable large carnivore species warrants a focus on interface areas and, in addition, requires evidence-based decision-making for sustainable management and policy development (*König et al., 2020*). An equilibrium in shared landscapes can only be achieved and maintained through the co-adaptations of carnivores and humans through a continuous and dynamic process (*Carter & Linnell, 2016*). To effectively manage the interface between tigers and humans, it is imperative to employ a dynamic interdisciplinary approach that continuously tests the effectiveness of location- and species-specific mitigation measures (*van Eeden et al., 2018*). Identifying and employing key functional mitigation measures is crucial in gaining local support and investment in policies that promote coexistence (*Karanth & Kudalkar, 2017*). With the rising tiger numbers around PAs in the country, our study emphasises the need to monitor and

manage the spill-over population in often sub-optimal interface areas. Albeit a smaller proportion, the animals at the interface can disproportionately affect local people's perception. Therefore, understanding their behavioural adaptations to negotiate human-dominated areas and their role in conflict dynamics would be vital in addressing interface issues. Incorporating this knowledge into village-level management and ESZ planning can help address economic prosperity while establishing a more robust support system for conservation efforts.

## ACKNOWLEDGEMENTS

MM would like to thank the Course Directors and Dealing Assistant of the Master's Programme; Dean and Director, Wildlife Institute of India, for the administerial support. MM expresses his gratitude to the Wildlife Institute of India faculties for their constructive comments and valuable suggestions; LEVL researchers–Dr Sankarshan Chaudhuri, Mr Supratim Dutta, Dr Meghna Bandyopadhyay and Mr R. Rajasekar for sharing their expertise in various aspects of data analysis, fieldwork and manuscript preparation. MM gratefully acknowledges the generous help and crucial suggestions from Dr Kees Mulder, author of the *circglmbayes* R package, in navigating the complexities of circular statistics. We thank Mr Shiv Pratap, Mr Arvind Raikwar, Mr Pappu Pal and Mr Amar Yadav for their dedicated efforts in field data collection and processing, Mr Darshan Singh for the logistical support and countless frontline staff of the Reserve in facilitating the fieldwork. Credits fall short of adequately acknowledging the immense contributions of the StackExchange community that directly and indirectly aided in the data analysis and visualisation aspects of this manuscript, the latter for which the assistance of OpenAI's DALL·E 2 was also utilised for aesthetic improvements.

### Funding

This study was financially supported by the dissertation grant of the Post-graduate Programme in Wildlife Science, Wildlife Institute of India (No. WII/AC/2019-21/ Dissertation/XVII M.Sc), and the project: 'Development of Landscape Management Plan and Monitoring with reference to Ken–Betwa River Link Project in Panna Tiger Reserve, Madhya Pradesh', funded by the National Water Development Agency, Government of India (No. WII/KR/PROJECT/PLMP/2017-18/F(1)). The funders had no role in study design, data collection and analysis, decision to publish, or preparation of the manuscript.

### Grant Disclosures

The following grant information was disclosed by the authors:
Wildlife Science, Wildlife Institute of India: WII/AC/2019-21/Dissertation/XVII M.Sc.
National Water Development Agency: WII/KR/PROJECT/PLMP/2017-18/F(1).

## Competing Interests

The authors declare that they have no competing interests.

## Author Contributions

- Manu Mohan conceived and designed the experiments, performed the experiments, analyzed the data, prepared figures and/or tables, authored or reviewed drafts of the article, conducted field work, and approved the final draft.
- Sambandam Sathyakumar conceived and designed the experiments, authored or reviewed drafts of the article, and approved the final draft.
- Ramesh Krishnamurthy conceived and designed the experiments, authored or reviewed drafts of the article, secured the funding and held the administrative role of the project, and approved the final draft.

## Animal Ethics

The following information was supplied relating to ethical approvals (*i.e.*, approving body and any reference numbers):

The GPS telemetry data used in the study were borrowed from the ongoing landscape project, as part of which two tigers were collared following strict animal ethics and protocols approved by the National Tiger Conservation Authority (NTCA), with permission from the Ministry of Environment, Forest and Climate Change (MoEFCC-Wildlife Division), Government of India (F. No. 1-19/2019 WL, dated 15/09/2020).

## Ethics

The following information was supplied relating to ethical approvals (*i.e.*, approving body and any reference numbers):

The research was reviewed and approved by the Faculty of Wildlife Sciences as part of the Post-graduate programme in Wildlife Science at the Wildlife Institute of India, Dehradun (WII/AC/2019-21/Dissertation/XVII M.Sc, dated 24/11/2020).

## Field Study Permissions

The following information was supplied relating to field study approvals (*i.e.*, approving body and any reference numbers):

Madhya Pradesh Forest Department (MPFD) issued all necessary permission for conducting fieldwork in Panna Tiger Reserve (Letter issued by the Principal Chief Conservator of Forests (Wildlife)/GEN./15-20/6043, dated 01/09/2018).

## Data Availability

Raw data are available in the Supplemental Files.

## Supplemental Information

Supplemental information for this article can be found online at http://dx.doi.org/10.7717/peerj.17693#supplemental-information.

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
