# Peer review of "Predator in proximity: how does a large carnivore respond to anthropogenic pressures at fine-scales? Implications for interface area management"

_PeerJ, doi:10.7717/peerj.17693_

## Round 0.1 · original submission · Major Revisions

Dear author,

Thank you for your submission and please forgive the delay in decision-making on my part.

I agree with the reviewers that the paper is interesting and will be a good addition to the literature. I also agree, however, that there are a number of issues that need to be addressed before the work can be published. Most notably:
1) The work requires considered editing to reduce its length;
2) I agree with reviewer 1 that making data available for review would be beneficial and I encourage open data where possible;
3) I agree with reviewer 2 that the use of occupancy models makes the included RAIs obsolete;
4) I also query the constancy of detection in those occupancy models;
5) Finally, I am not convinced by the 1km distance applied to activity/overlap analyses and would like to see this either robustly justified or adjusted appropriately.

I look forward to reading your revised submission.

Best wishes,
Anthony

Reviewer 1 ·

Basic reporting

Overall, this is a well-written manuscript with interesting questions and findings. However, certain sections of the manuscript need to be significantly shortened to improve flow and coherence and to retain reader interest. Specific comments:

- While most of the manuscript is well-written and appears to have been reviewed for language and writing issues, the abstract needs several minor edits

- Lines 37-38: The phrase ‘This suggested that the tigers.. anthropogenic influences’ is worded a bit strangely. While habitat exploitation by tigers can be influence by anthropogenic influences, this may have nothing to do with their ability to do so – they just seem to prefer not doing that

- Line 44: The meaning of the phrase ‘inversely related to depredation probability’ is unclear in this context

- The Introduction section is extremely long and, in some instances, circles back to topics discussed in previous paragraphs. This section needs to be significantly edited and restructured. For example, the current paragraph themes are as below:
i. Large carnivore habitat loss and threats
ii. How large carnivores tend to adapt to these threats
iii. Impacts of large carnivores on humans
iv. Carnivore behavior in human-dominated areas (a bit of an overlap with ii, can be combined)
v. Carnivore recovery – this is very long and can be reduced to 1-2 sentences and merged with paragraph 1. Most of this content is tangentially relevant to the analysis
vi. Human-tiger conflict (again related to the current paragraph 3 – needs to be combined)
vii. Issues and challenges in the Central Indian region – this is also too long and can be merged with the conflict and problems caused by large carnivores paragraph

- Lines 160-166: This content should be moved to the methods section.

Figure 1: This figure would be more ‘readable’ with a less busy base layer for the map. It is also unclear what the acronyms (TOF) and letters/numbers in the legend mean – these should be clarified or removed. I also cannot see any Northern Dry Mixed Deciduous Forest (pink box) in the map.

Figure 2 & 4: Move to supplementary data

Figure 5: Same comment as Fig 1: a cleaner base map layer would improve this figure

Raw data does not seem to be available for download in the files available for review – there are a bunch of summary tables and results, but not the raw data. Please explain if you have an exemption from releasing raw data due to the potential for misuse.

Experimental design

The research questions are well-defined, relevant and meaningful, and the analysis seems to be rigorous. I have 2 minor comments for this section:

- Line 382: I would suggest also including dusk and dawn as additional categories, as many species behave very differently during these periods. Additionally, it would be more robust to use actual sunrise, sunset, dawn and dusk timings given the variation in these. Several R packages make this fairly straightforward.

Line 473: Home ranges are not previously mentioned in the methods section. Although this is not a major area of emphasis in this analysis, using a more robust home-range calculation methodology (for example, AKDE) may be appropriate.

Validity of the findings

Underlying data have not been provided; consequently, there is limited scope to evaluate the validity of the results. The Discussion section is well-written and linked well to the original questions and reported results. Specific comments:

- Lines 541-542: This statement needs to be supported with stronger data/a publication, rather than a news report on a single incident. I understand that highway mortalities are indeed a significant problem for large carnivores in India, and a stronger reference would bolster this assertion.

Section 4.4: This is an extremely long sub-section that needs significant editing.

Additional comments

This is a well-written manuscript based on excellent science that makes for interesting reading, and would be a good addition to PeerJ. I recommend its acceptance after the authors have addressed the issues listed above.

Reviewer 2 ·

Basic reporting

Thank you for the opportunity to review the manuscript “Predator in Proximity.” The manuscript includes a great deal of field work and analysis of human-tiger interactions. I applaud the authors for taking a human-tiger perspective to this work instead of examining each component independently. I think the manuscript could make a nice contribution once several key issues are addressed.

1. The Introduction and Discussion are too long. They both tend to drown the main points in literature cited. For example, the Introduction can be cut by at least 1 page by assuming that readers already understand the challenges of carnivore conservation in human-dominated areas. The authors should present the key knowledge gap they intend to fill by the 2nd paragraph. Rather there is a lot of text listing threats to carnivores, the importance of mitigating conflict, etc. In the Discussion, please summarize the key insights from this work in the first paragraph and then explain the importance of each insight in subsequent paragraphs. There is a tendency to refer to all other studies that confirm or contradict each finding from this study, and that makes it challenging to determine the main message the authors would like the reader to take away. For example, that tigers are less nocturnal when near villages strikes me as counterintuitive given the research on how wildlife are more nocturnal under greater human disturbance. The authors should laser-focus on the novelty of each key result in the context of prevailing ideas about wildlife use of human-modified areas.

2. Regarding the novelty, I think the key point of the ms is that livestock (and human activities) impact tiger space use, and that tiger space use might influence the likelihood of livestock conflict (when accounting for human presence such distance to human villages and roads). The common denominator thus is tiger space use. I would elevate that point in the Introduction and explain why doing both analyses jointly provides insights that would not be gained from doing the 2 analyses separately.

3. Related to the GPS telemetry, I think the authors should more clearly describe what they hope the GPS telemetry provides in addition to the camera trap data. The core analysis involves the camera trap data in my opinion. The GPS telemetry provides validation? Or is it simply ancillary information on spatio-temporal movements that are at finer spatial and temporal resolutions than the CT data? Another way to think about it, does adding the GPS telemetry change the results (or authors’ interpretation of the results) than if they were not included?

4. The paper suffers a little from a “chicken and the egg” problem. That is, do tigers respond to livestock/human activities or do they behave in ways to hunt livestock more effectively? Likely tigers are seeking to largely avoid human disturbance; however, they might occasionally encounter livestock and take the opportunity to kill them. Distinguishing the order of things is important because otherwise the authors are contradicting themselves with results suggesting avoidance on one hand, but then showing that there are spatial patterns of livestock depredation (suggesting attraction of tigers to those areas to kill livestock).

5. Much of the text in the Results belong in the Methods. For example, L. 425-437, L. 439-443. These all describe the variables and modeling approaches.

Experimental design

1. It seems the authors use both RAI and occupancy outputs for tiger space use. However, I have a couple issues around these. First, I’m not sure why detection was held constant in the occupancy models. An intercept-only detection model is usually considered the null model and without comparing model fit to other models that include detection covariates, it’s hard to justify not including detection covariates. Not to mention, keeping detection constant diminishes the utility of the occupancy framework in which you can examine the role of covariates on both occupancy and detection processes. I’d like to see the authors include some detection covariates as well. Second, I do not see the value of doing an additional analysis of RAI. Occupancy is a better measure of space use and RAI is increasingly viewed an obsolete since it does not include the detection process. And as the authors have already conducted the occupancy analysis I do not see the added value of the RAI—an inferior metric. I recommend the authors delete the RAI portions and replace with tiger occupancy. I realize that this will involve re-doing some of the other livestock depredation models but I think doing so will be more robust.

2. I’m not convinced conducting a temporal activity/overlap analysis by 1km distance bins makes sense for tigers. They traverse several kilometers easily in a single day. I find it hard to justify that their diel activity patterns would demonstrably shift when 1km away since many of those same tigers are occurring across multiple distance bins, right? Can the authors determine that the tigers <1 km are totally different from the tigers at 1-2km, and so on? Instead, I think looking at the proportion of night/daytime locations from the GPS telemetry when near the villages compared to far away is more justifiable (and is already in the text).

3. Why not include distance from villages as a continuous variable instead of a categorical variable with arbitrary binning? The continuous variable retains more information and can still be visualized easily as a marginal effect (i.e., examine the effect of distance on tiger occupancy while keeping all other variables at their mean).

Validity of the findings

No comment

---

## Round 0.2 · Minor Revisions

Dear authors,

Thank you for your resubmission and considered revisions. The reviewers have requested a few additional amendments but these are very minor and can be dealt with very quickly. Unless you decide to make substantial changes for some reason, I do not anticipate the revised manuscript being sent for further review.

Best,
Anthony

Reviewer 1 ·

Basic reporting

I have a few minor comments for this version of the manuscript:

Line 21-22: Might be good to preface this with a broad opening statement, for example: Habitat loss is one of the major threats to large carnivore conservation, often leading to increased movement of large carnivores in human-dominated landscapes.

Line 22-23: This is a slightly strangely phrased sentence that could be made clearer – carnivores face increased energetic costs, restrict their activity in these landscapes and face higher persecution risk.

Lines 26-27: Critical tiger habitat and protected area should not be capitalized.

Lines 43-44: Two of the factors discussed here have p>0.05. As a non-significant result (assuming a threshold of 0.05), these should not be listed here.

Lines 122-123: The sentence ‘This query is..’ could be simplified. Also, adding ‘the’ before general perception and tiger might be appropriate (or tiger could be changed to tigers).

Lines 123-124: Remove the phrase ‘taking the objective criteria’. This sentence could be framed as ‘Our study introduces..’

Line 133: There is a missing ‘o’ in anthropogenic

Line 138: Remove the phrase ‘in their temporal activity’ as it is superfluous

Lined 143-144: What does ‘reconciliation with ensuing conflicts’ mean?

Line 483 and 485: Remove the word ‘the’ at the start of the sentence ‘The tigers..’

Lines 519-521: This is an awkwardly framed sentence. I am not sure what the ‘it’ here refers to.

Experimental design

No comments

Validity of the findings

No comments

Additional comments

The authors seem to have addressed most of the suggestions made/issues highlighted in my first review of this manuscript. My only comments in this round of review comprise minor issues around specific lines in the manuscript. I recommend the acceptance of this manuscript after the authors have addressed these.

Reviewer 2 ·

Basic reporting

I applaud the authors for their revisions which addressed all my comments. The manuscript is a very nice contribution to the literature. Some additional minor comments below.

Minor comments:
L. 122-127 from “This query is deviant…” to “location-specific management interventions for tiger conservation” can be deleted.

L. 137: change to “, such that tigers would be less likely to use areas near to human settlements.”

L. 743: revise to “…avoidance of human settlements and infrastructure (Sarkar et al. 2017; Carter et al. 2023)”
Carter, N.H., Zuckerwise, A., Pradhan, N.M.B., Subedi, N., Lamichhane, B.R., Hengaju, K.D., Acharya, H.B. and Kandel, R.C., 2023. Rapid behavioral responses of endangered tigers to major roads during COVID-19 lockdown. Global Ecology and Conservation, 42, p.e02388.

Sincerely,
Neil Carter

Experimental design

NA

Validity of the findings

NA

Additional comments

NA

---

## Round 0.3 · Minor Revisions

Dear author,

Thank you for your revision. I am happy with your amendments, with one extremely minor but I believe important caveat. While I agree with you about the arbitrary nature of P = 0.05 and the importance of considering all variables included within the best approximating model, I do not agree with the statement in the Abstract that the variable "had weak statistical significance." If we are to use the threshold at any point in our work, however, then findings are either significant or not. 'Weak' significance and other similarly vague terms (e.g. 'approaching significance', 'trend towards significance') are inappropriate in my opinion. I would appreciate it if you would amend that one part of the sentence to 'non-signficant'. Everything else about your inclusion of those variables and the way in which they are contextualised is absolutely fine.

Best,
Anthony

---

## Round 0.4 · accepted · Accept

Thank you for your prompt response. I am happy to recommend that your submission be accepted for publication.